# "It's my calling", Canadian dog rescuers' motives and experiences for engaging in international dog rescue efforts

Kai Alain von Rentzell[1]*, Christiana Bratiotis[2], Alexandra Protopopova[1]

1 The Animal Welfare Program, Faculty of Land and Food Systems, University of British Columbia, British Columbia, Vancouver, Canada, 2 School of Social Work, University of British Columbia, British Columbia, Vancouver, Canada

* kaialainv@gmail.com

## Abstract

The importation of rescue dogs has become an increasingly common occurrence in recent years, often involving industrialized countries as the ultimate destination. However, international dog rescue activities have attracted considerable criticism from the veterinary community and the public due to the associated zoonotic disease and public health risks, as well as the potential poor behaviour of international rescue dogs. The Government of Canada has also recently placed a temporary suspension on all commercial dog imports from non-rabies free countries due to the growing concerns of the zoonotic disease risks. To understand the perspectives and experiences of stakeholders involved in dog import activities in Canada, we interviewed nine members in leadership positions of Canadian-based international dog rescue organizations. Thematic analysis of interview dialogue yielded three themes: 1) *Motive*, which described the reason behind participants' involvement in international dog rescue; 2) *Challenge*, which described the major difficulties faced in participants' dog rescue work; 3) *Duty*, which described participants' beliefs on responsible dog rescue practices. Members of international dog rescue organizations described being driven by strong desires to provide animal and humanitarian aid. However, local dog rescue efforts were constrained by logistical and societal barriers unique to the Canadian context. Additionally, the current study revealed both similarities and differences in occupational experiences between international dog rescue organizations and other animal care professions. Specifically, difficulties with the emotional burden associated with caregiving professions was also present within dog rescue work. However, international dog rescue members also experienced additional challenges due to the stigma surrounding international dog rescue operations. Further research on attitudes held by other stakeholders involved in dog import activities, as well as members of Canadian communities needing dog rescue aid may provide meaningful inputs on how to better support and facilitate local and international dog rescue efforts.

**Data Availability Statement:** All relevant data are within the paper and its supporting information files.

**Funding:** The study was funded in part by the Natural Sciences and Engineering Research Council of Canada (www.nserc-crsng.gc.ca) and the British Society for the Prevention of Cruelty to Animals (https://spca.bc.ca/) through a grant to AP as an Industrial Research Chair in Animal Welfare (No. 554745-19). The funders had no role in study design, data collection and analysis, decision to publish, or preparation of the manuscript.

**Competing interests:** The authors have declared that no competing interest exist.

## Introduction

Canada is home to over 8 million dogs, with over 40% of households owning at least one dog [1]. The average lifespan of companion dogs is reported to be 11 years [2, 3], suggesting that the annual demand for dogs in Canada is roughly 727,000 (see Norman et al, 2020). Despite a large demand for dogs, roughly 10%, or approximately 3,000, of Canadian shelter dogs are euthanized each year [4–6]. Furthermore, northern communities in Canada continue to experience challenges with free-roaming dog populations, thus frequently resorting to culling events due to lack of alternative humane interventions [7–9]. Even though there may be sufficient numbers of Canadian dogs to meet the demand, the Canadian National Canine Importation Working Group reported that at least 6,189 rescue dogs were imported to Canada in 2013 [10]. These reports seem to illustrate a perplexing scene in the Canadian sheltering practices, where many adopters are choosing to adopt from international communities rather than from local shelters despite apparent dog rescue needs in Canada.

Dog importation has become an increasingly common phenomenon globally, with many efforts involving rescue dogs [11–15]. Despite their ubiquity, international rescue organizations have been the topic of much criticism from public health experts given the risks associated with the prevalence of disease in dogs coming from unknown origins [13, 14]. Canine rabies is the most frequently mentioned disease due to its severe animal and public health risks [10, 13, 16, 17], although other zoonotic diseases such as alveolar echinococcosis and leishmaniosis also present public health risks, and dog importation is the most likely route for the spread of disease to novel areas [18–20].

Potential poor behaviour of international rescue dogs is also a concern raised by veterinarians and animal welfare proponents [14, 21]. And certainly, studies show that early-life experiences in dogs, particularly during their socialization period (3–14 weeks of age) can influence their subsequent ability to cope with a variety of social and environmental stimuli (e.g., unfamiliar dogs and people, traffic, and household appliances) [22, 23]. Internationally-sourced rescue dogs, with their diverse backgrounds can have varying experiences with people from a young age, which range anywhere from positive to negative. For example, free-roaming dogs are often provided with food by local people [22, 24], but also experience high mortality rates due to human-mediated effects [24, 25]. These interactions can give rise to dogs with varying associations with people; some free-roaming dogs may develop a fear or aversion towards people while others may be well-socialized and adapted to life in human-dominated contexts. While research on the behaviour of rescue dogs is limited, currently available evidence suggests that prevalence of extreme behaviours in internationally-sourced rescue dogs is comparable to dogs originating from other sources [11, 14, 26].

Reports of dogs arriving at international airports with inaccurate medical records have increased over recent years [11, 17, 27–29], no doubt exacerbating the growing suspicions of illegal dog importation activities (i.e., puppy smuggling). Organized illegal dog importation is a major concern, since importers profit at the expense of dog's health and welfare [29]. For example, illegal importers may deliberately traffic underage dogs (<4 months old) since they can be sold at a high price [28, 29]. Dogs this young may not be physically fit to endure long international flights, and may be at greater risk of zoonotic diseases due to premature immune systems [13, 29]. Unfortunately, this may incite further skepticism towards responsible dog rescue organizations that import dogs. While the prevalence of illegal dog importation in North Americas is currently not well understood, researchers have suggested that this may be a developing problem [27, 29].

Despite concerns associated with dog importation, hundreds of dog rescue organizations continue to import dogs to Canada from international communities [10]. As a result,

international dog rescue activities have become a topic of heated debate. While the attitudes of the general public [11, 14, 15] and veterinary professionals [10, 14, 30] towards dog importation have received some attention by the scientific community, perspectives of dog rescue organizations that engage in international rescue have, to our knowledge, never been investigated. It is now generally accepted that attitudes and perceptions held by the public and wider society can play a major role in government policy decisions on topics surrounding animal welfare [31, 32]. Thus, understanding the views of individuals working within the dog rescue organizational framework can potentially provide valuable insights into the current state of Canadian dog rescue and sheltering practices. Therefore, the current study aimed to explore the unique perspectives, experiences, and the "subjective realities" of members of international dog rescue organizations based in Canada through a reflexive thematic analysis of participant interview dialogue [33]. Specifically, the current study aimed to understand the motives of rescue organizations that import rescue dogs to Canada, and to explore common challenges they faced in their line of work.

## Methods

### Participants

The study was approved by the UBC Behavioural Research Ethics Board (H21-02328). Participants (n = 9) were recruited using a combination of purposive and snowball sampling methods to ensure the retrieval of rich information on the research topic [34, 35]. To be eligible for participation, individuals had to meet two criteria: 1) be a member of a Canadian dog rescue organization involved in international rescue and importation; and 2) hold a managerial, administrative, or a similar leadership position of their rescue organization. The recruitment began with a general search of dog rescue organizations through Petfinder.com using the keyword "Rescue" and British Columbia (BC) set as the location. While BC was chosen as the location for initial research interest in international dog rescue activities in this region, this decision was later amended to include rescue organizations based in other provinces due to difficulty recruiting BC-based participants, which resulted in the recruitment of rescue organizations based in Ontario, Canada (ON). Online websites of all identified rescue organizations were accessed to determine whether they engaged in international import of dogs prior to any contact being initiated by the research team. No contact was made for rescue organizations that did not import dogs or if this information was unclear.

A total of 55 rescue organizations were contacted by email between May 2021 and March 2022. The email contained a letter of initial contact (S1 File), which was constructed prior to the participant recruitment period. Whenever a rescue organization responded to the recruitment email with interest in participation, a follow-up email containing the interview consent form was sent to acquire written consent prior to scheduling the interview. At the end of each interview session, participants were asked for referral contacts of other rescue organizations that may be interested in participating (i.e., snowball sampling) that resulted in a single additional participant. A total of nine participants agreed to take part in the interview. All participants were in managerial or similar leadership positions of dog rescue organizations based in Canada that regularly import dogs from other countries. Some characteristics of the participating rescue organizations are included in Table 1.

### Research team and positionality statement

The primary investigator (Kai von Rentzell) is a master's student enrolled in the Animal Welfare Graduate Program. von Rentzell's research home is within the Animal Welfare Program, which is a research group located in the Faculty of Land and Food Systems at The University

**Table 1. Participant information.** Descriptive information of participants, rescue base province, participant occupational position, primary dog source location, and types of animals rescued by their rescue organization.

| Participant ID | Rescue Province | Participant position | Primary dog source locations[a] | Animals rescued |
|---|---|---|---|---|
| P1 | BC | Regional Coordinator | South Korea<br>Mexico<br>China<br>Iran | Dogs and Cats |
| P2 | ON | Media Manager | Mexico<br>Dominican Republic<br>Canada (Manitoba)<br>US (California, Texas, Tennessee)<br>Egypt<br>Cayman Islands<br>Barbados<br>Costa Rica | Dogs |
| P3 | BC | Founder | US (California)<br>Taiwan<br>Mexico<br>Lebanon<br>Afghanistan[b]<br>Romania<br>Saudi Arabia<br>Canada | Dogs and Cats |
| P4 | BC | Founder | Mexico | Dogs |
| P5 | BC | Founder | US (Alabama) | Dogs |
| P6 | BC | Founder | Mexico<br>US (California)<br>Canada | Dogs |
| P7 | BC | Founder | South Korea<br>China | Dogs |
| P8 | ON | Rescue Board Member | Mexico<br>Saint Lucia<br>Virgin Islands<br>Columbia<br>Dominican Republic<br>Costa Rica<br>Jamaica<br>Bahrain<br>Egypt<br>Nepal | Dogs |
| P9 | ON | Founder | Canada (Northern Ontario, Manitoba)<br>US<br>South Korea<br>Nepal<br>Kuwait<br>Iran<br>Mexico<br>Ukraine[b] | Dogs |

a. Primary rescue area is not an exhaustive list of all dog rescue regions. Rather, it is a list of regions from which rescue members recalled during the interview session.
b. Countries from which rescue operations were planned, but not yet executed.

of British Columbia (UBC). von Rentzell was born and raised in Japan and has a Japanese and German ethnic background. von Rentzell grew up with a personal interest for animals, which led him to pursue formal education in animal welfare through his undergraduate degree in Applied Animal Biology also at UBC. von Rentzell have also practiced companion animal care in Japan through his work as an animal caretaker at a veterinary clinic. von Rentzell has also volunteered in a dog rescue organization to better understand the rescuer's "insider"

perspective. However, his "western" education led him to be more familiar with pet keeping practices in Canada and Japan. von Rentzell's goal in his research is to explore the topic of dog importation through unique perspective of those, who engage in international rescue, to identify ways of improving import practices without completely banning international rescue.

The current study was overseen by Dr. Alexandra Protopopova, an assistant professor in the Animal Welfare Program at UBC and a Certified Applied Animal Behaviorist. Protopopova has an extensive professional and academic background in animal sheltering, behaviour analysis, and companion animal welfare. Protopopova has worked intensively with colleagues in animal shelters in Florida and Texas, US, which highlighted for her the overpopulation of animals in shelters as well as the difficult work that animal shelter staff must carry out. Therefore, Protopopova's goals are to assist animal shelter workers in the southern part of the US, and identify whether importation can be used to move animals to more adoptable areas without sacrificing animal welfare.

Dr. Christiana Bratiotis is an Associate Professor in the School of Social Work at UBC. She is a leading researcher on community-based approaches to hoarding behaviour and the development of cross-sectoral harm reduction assessment and intervention practices. Bratiotis has expertise in qualitative research methodologies, and has used qualitative inquiry to explore the experiences of people who hoard during a discarding task [36], as well as the emergence, perspectives and practices of hoarding task forces [37], among others [38].

As illustrated through the positionality statements, the research team consist of members in academia, one of whom is directly engaged in international dog rescue activities, which allows perspectives from both an "outsider" or an "etic" perspective as well as an "insider", "emic" perspective [34, 39]. This is a crucial component to recognize as the researchers are the instrument in qualitative research, and their unique backgrounds and cultural perspectives give rise to the research interests and also potential biases in data collection, analysis and reporting [34]. To address the potential for bias and increase the trustworthiness of the data, the current researchers engaged in practices to increase the qualitative rigor [40]. For example, the unique positionalities of the research team was critically reflected by "memo-ing" the thoughts and insights that occurred during the analytic process (i.e., reflective memos) [34]. While the researchers engaged in critical self-reflection, the unique positionalities of the research team were also embraced to better understand rescue operations from both an insider and outsider perspective. Additional steps to ensure qualitative rigor can be found in the Data Analysis section.

## Data collection

Data were collected between September 2021 and March 2022, which coincided with the participant recruitment period. Data were collected through individual dialogue between the first author and the representative member of a dog rescue organization in a semi-structured interview format. All interviews were conducted by the first author, hosted online through Zoom Video Communications, and were up to 1 hr in duration. All interviews were audio recorded to preserve the accuracy of the content until transcription was completed. Semi-structured interviews were used to generate an abundance of rich, in-depth account of the participants' lived experiences [34].

An interview guide was used to navigate general topics of interest during the interview, but not the exact course [41]. The interview guide contained six open-ended questions designed to explore the lived experiences of rescue members. Apart from the final question, each question had one or more planned follow-up questions for further expansion on the topic. Additionally, the semi-structured nature of the interview allowed for the investigator to ask additional

**Table 2. Interview questions and planned follow-up questions.**

| |
|---|
| **Question 1:** Tell me about how your organization came to the decision to rescue dogs from [geographic area]? |
| **Follow-up:** Please elaborate on how you got involved with these shelters in these source communities? |
| **Question 2:** Tell me about the factors you consider when selecting dogs to bring over from [geographic area]? |
| **Follow-up:** Please describe to me what life is like for a dog before they are rescued? |
| **Follow-up:** Please describe to me a "typical" dog you rescue? |
| **Follow-up:** From talking to other rescues, they often have a procedure of finding the "right match" between the dog and the owner. Does your rescue also have that, and please elaborate on that? |
| **Question 3:** What are the costs that are involved with rescuing dogs from abroad? |
| **Follow-up:** How do you recuperate these costs? |
| **Follow-up:** Has there ever been a time where you couldn't rescue a dog because it was too expensive? |
| **Question 4:** Tell me about some challenges you face through your work? |
| **Follow-up:** What can be done to reduce or eliminate these challenges? |
| **Question 5:** Please tell me what it is that keeps you continuing your work? |
| **Follow-up:** In the perfect world, how would the lives of all dogs look for you? |
| **Follow-up:** What do you enjoy the most in your work? |
| **Question 6:** Before I conclude this interview, is there anything I haven't asked you about that you think is important for me to know? |

unplanned follow-up questions and clarifications on topics discussed during the interview [34]. Interview questions and their associated planned follow-up questions are shown in Table 2.

At the start of every interview session, participants were reminded of the study aim and their right to terminate the interview process at any time without any consequences. This introductory statement is also available in the interview guide (S2 File). Participant's verbal consent to be audio recorded was obtained prior to starting the recording and the interview.

Eight out of nine interviews were transcribed by the first author. The nineth interview was transcribed by an undergraduate research assistant, with the accuracy of the transcribed material checked by the first author. All interview dialogue were transcribed without the use or assistance from transcription services or software. Transcribed and de-identified interview dialogue was constructed in an online Microsoft Word document. The interview transcripts, redacted to ensure anonymity of the participants, are available in the supplementary materials (S3 File).

## Data analysis

Using the phenomenological approach guided by a social constructionist epistemology [42, 43], the current study aimed to explore the unique perspectives, experiences, and the "subjective realities" of members of international dog rescue organizations based in Canada. Interview transcripts were used as raw data for a reflexive thematic analysis [33]. The data analysis followed the steps outlined by Braun and Clarke (2006) and Nowell et al. (2017): 1) familiarization of the data; 2) generation of initial codes; 3) searching for themes; 4) reviewing themes; 5) defining and naming themes; and 6) production of the final report. Importantly, data analysis commenced in an iterative format; whereby, the analytic process reverted back and forth between phases [44].

By way of constructionist epistemology, the focus of the current study was on the latent content (i.e., underlying meanings and ideas) of participant dialogue [45, 46]. As such, *codes* were defined as meaning units (words, sentences, or paragraphs) that highlight the participant's experience with the phenomenon of international dog rescue [46, 47]. Codes were

aggregated into *Themes*, which are multi-faceted, abstract ideas that capture important aspects of the data, which appears in various degrees of patterned response or meanings [45]. A thematic network was used for the conceptualization of each theme to refine its structure of participant's experiences [44, 48]. Themes were identified inductively, such that themes were generated without an attempt to fit it into existing literatures, frameworks, or theories [34, 44, 45].

All coding of the raw data and initial development of themes was conducted by the first author. All codes were hand-coded using the following electronic apps: 1) Microsoft Word for initial identification of codes; 2) Google Docs for the construction of a codebook containing codes, reflexive memo, and the corresponding themes; 3) Lucid Chart (Lucid Software Inc.) for the development of the thematic network. There were no use of additional electronic apps or coding software to assist with the data analysis. Steps 1 and 2 were conducted entirely by the first author. However, the refinement of the nature of each theme, as well as the development of the thematic network was conducted through weekly supervisor debriefing until sufficient conceptualization of participant experiences was achieved [34]. The saturation of the data was assessed through a recursive process on the basis of practicality (i.e., time, cost, and effort) of conducting successive participant recruitment and interview, as well as on the level of richness, diversity, and depth of participants' experiences captured through the analysis in addressing the research questions [49, 50]. Additionally, all interview transcripts with identifiable information removed are available in the supplementary materials (S3 File).

During the data analysis, we found that participant expressions for the theme, *Motive*, frequently reflected their understanding of the real-world issues surrounding international dog rescue operations. While findings are presented thematically, participants' conceptualization of their worldview (named Descriptive insight) is presented jointly with the theme *Motive* to better illustrate participants' experiences and understanding of international dog rescue operations.

## Results and discussion

Findings of the current study are presented as three Themes: *Motive*, *Challenge*, and *Duty*, each containing multiple sub-themes (Table 3). While *Motive* and *Challenge* are aligned with general topics conceptualized during the construction of the interview questions, the analysis resulted in an additional theme, *Duty* which was not conceptualized during the research

**Table 3. Themes, sub-themes, and sub-sub-themes.** Themes with relevant sub-themes and corresponding sub-sub-themes.

| Themes | Sub-themes | Sub-sub-themes |
|---|---|---|
| Motive | Desire to help | N/A |
| | Barrier | N/A |
| | Help Canadian adopter | N/A |
| Challenge | Adopter expectation | N/A |
| | Emotional stress | N/A |
| | Rescue stigma | N/A |
| | Finances | N/A |
| Duty | Owner-dog relationship | Dog assessment Honesty Eternal safety net |
| | Canadian society | Thorough medical screening Specific rescue dog criteria |
| | Rescue partner | N/A |

design phase. The final theme emerged from the data through the nature of inductive thematic analysis.

## Motive

**Descriptive insight.** Within the theme *Motive*, we captured the rescue member's reasoning and experiences that led to their involvement in international dog rescue operations. Interestingly, the thematic network revealed that rescue member's rationale for getting involved in international dog rescue also reflected their conceptualization of the greater structure of dog rescue in general (Fig 1).

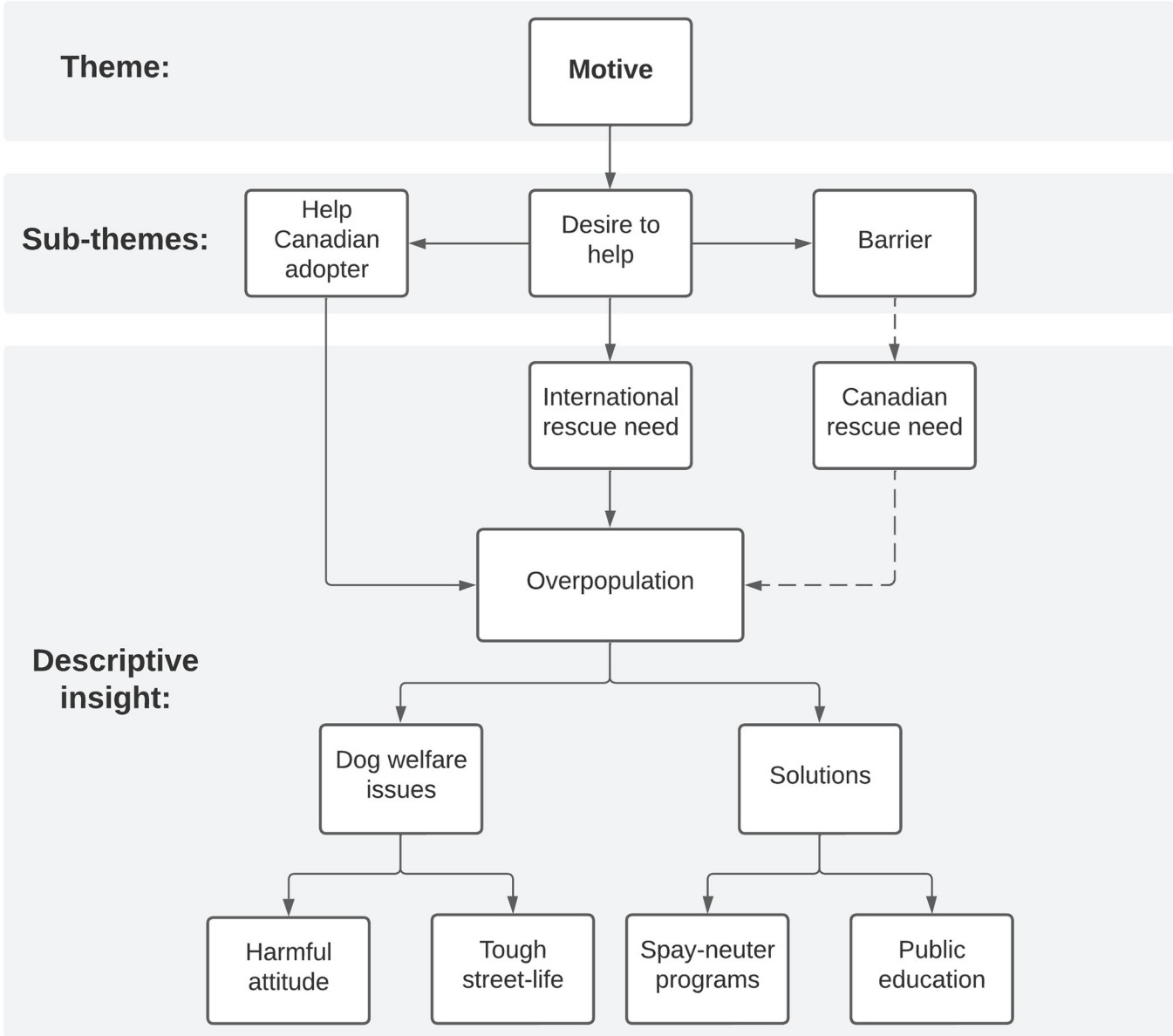

**Fig 1. Thematic network of *Motive* and rescuer worldview.** Thematic network of *Motive*, its sub-themes, and its relation to Canadian-based international dog rescue member's worldview on dog rescue efforts.

In their description of their worldview regarding dog rescue, participants identified the need for dog rescue operations in both local and international communities. In either community, the root cause for dog-related issues was identified as dog overpopulation. These descriptions are consistent with existing literature that suggest various dog-related issues as a consequence of dog overpopulation, although what is considered "too many dogs" vary for each community [8, 9, 24]. Participants linked the continuity of dog overpopulation to a lack of government involvement and a lack of financial or veterinary resources to tackle the issue with long-term results. These perspectives are also supported by previous literature where lack of government support and lack of financial or veterinary resources were major setbacks in areas facing free-roaming dog population challenges [8, 51]. P9 described their perspective in detail:

*We do have reserves [in Canada] and the communities up north that do and conduct culls because they don't have the resources to spay and neuter their dogs [...] So, the overpopulation, which is the main problem anywhere in Canada and abroad, is what brings this [issue] up. The way that the [Canadian] communities [...] deal with this with this is by culling, right? [...] In the US, they have kill-shelters where they euthanize for space [...] but the main thing is that there's an overpopulation of dogs and that they're getting killed right? And that is not resolving the problem. (P9)*

Overpopulation of dogs resulted in two broad types of dog welfare concerns: Poor public image of dogs, and Tough street-life conditions. Poor public image of dogs manifested in various forms. In extreme cases, poor public image of dogs resulted in physical abuse, which participants identified as a clear compromise to the welfare of dogs. P8 described how harmful labelling reinforced some international community member's disregard towards the dog's welfare.

*When I think of something like Mexico or poor area of Mexico, [the dogs are] considered "vermin", right? You run it over, you run it over. They get hit by a rock, they get hit by a rock [...] So, it's just stuff that would never fly here [in Canada] is happening out there. (P8)*

A more subtle consequence of poor public image was reflected in the lack of public interest for shelter dog adoptions. P1 identified that a lack of public interest for adoption in the country of origin contributed to the flood of rescue requests their organization received regularly from international communities.

*We have far more people contacting us asking if we can help them, than we are able to help. A lot of them have no other place they can send these dogs. [Dogs] can't get adopted in their country of origin, [the public are] just not interested, and it's very hard for [international rescues] to find anywhere that will accept [dogs]. So, they reach out to us and other organizations similar to us. (P1)*

However, several participants made explicit mention that challenges with dog overpopulation were not always a consequence of poor image and harm inflicted by the local people. Participants described how free-roaming dogs commonly faced tough street-life conditions and were exposed to various dangers associated with the street environment. Overpopulation exacerbated these risks through increased competition for resources between dogs and amplified risk of exposure to diseases. P4 described the range of difficulties the free-roaming dog population regularly encountered.

*Well, it's a life probably running around looking for something to eat, and in the elements. Probably being shooed away by somebody so that they're not in the way [. . .] But yeah, there's a lot of risks on the road, on the streets for dogs. And there's health dangers, there's dangers of other dogs, being attacked by other dogs, and then there's, you know there's overpopulation [. . .] There are also diseases that they can get from running free. (P4)*

Throughout the interview sessions, two interventions were emphasized as necessary elements in achieving long-lasting changes with dog overpopulation issues: large-scale spay and neuter programs, and public education. P8 expressed that international rescue efforts themselves were not enough to resolve overpopulation challenges. Instead, they expressed a greater goal of promoting networks between animal welfare groups and increasing awareness in international areas to facilitate future connections and incentivize greater aid for animal care efforts.

*Initially when I started doing [international rescue], we were trying to raise awareness in those [areas], for those areas. [. . .] It wasn't just [. . .] plucking that one dog out and bringing it up because we're never going to. . . you know, we're not going to rescue all of them. It's not going to happen in our lifetime. But the education and sterilizing these dogs especially when they have nowhere to go, [is] critical. And I know we do that here, but we need to put more effort and feet on the ground and finances and controlling populations elsewhere. So that's a big part of why I do [international rescue]. Doesn't mean I raise money for them to do those clinics but more awareness so people can connect and donate and get these things going. It's still a long way away [. . .] but, if I can provide support in that way, that's just [through] education. (P8)*

Free-roaming dog populations are found in various communities across the world, but the dog population dynamics, relationships with human members of the community, and the issues resulting from dog overpopulation differ for each community [52]. The participants' worldviews illustrate the different types of issues that can result from dog overpopulation.

Participants emphasized the sterilization of dogs as the most preferred approach for addressing dog overpopulation issues. Researchers support this idea, as controlling the free-roaming dog population through reduced reproduction rather than increased mortality is preferred in most scenarios as it is more humane and will likely result in greater community support [53]. Public education was another intervention suggested by participants, which is also supported by the scientific literature. Previous work suggests that most free-roaming dogs rely on humans to varying degrees for their survival, and self-sustaining population of feral dogs (having no reliance or associations with people) is generally rare [24, 53, 54]. Moreover, many free-roaming dogs are actually owned dogs that are not confined, meaning that engagement from dog owners to partake in husbandry practices that does not perpetuate dog overpopulation issues is pertinent [52, 53].

However, researchers also emphasize the importance of conducting localized research to assess the attitudes, perspectives, and needs from local community members to design an intervention that can effectively address the primary concerns of the given community [9, 52, 53, 55]. As participants recognized, international dog rescue efforts alone are likely not a solution to dog overpopulation. However, community engagement, as participants described being involved in, is likely to have a positive impact on the long-term dog population management and animal welfare outcomes [52]. These findings illustrate that some international dog rescue organizations are not only an animal relocation program, but also strive for positive animal welfare outcomes through involvement in various forms of community engagement.

**Desire to help.**   All participants expressed a desire to help as a core motive for their continued voluntary efforts. The participants were all highly committed to dog rescuing and engaged in the act of dog rescue out of good will. Indeed, for some participants dog rescuing carried a particularly strong existential meaning. For instance, P3 spoke about early childhood experiences caring for animals through their parent's involvement in animal-related work. For them, dog rescuing was not just volunteer work, but rather their "purpose in life".

*So, for me, it's passion. If I didn't truly love helping animals and feel like this was my calling in life regardless of getting paid or the status. I don't care about all of that. For me, to the core of my heart, I want to help animals for the rest of my life, and I truly know that [. . .] I feel lucky because I think I've found my true purpose in life. (P3)*

Unsurprisingly, participants' desire to help was commonly expressed in the context of helping dogs. Participants shared the belief that international dog rescue organizations are saving the lives of dogs that would not have otherwise survived. These beliefs were often expressed when describing reasons for continuing to volunteer in a dog rescue organization. For example, P1 expressed:

*Just seeing the transformation in the dog's lives, you know, from [. . .] facing certain death to running along a beach in British Columbia and snuggling in bed with people, I mean the contrast is so huge and that's really rewarding. (P1)*

In line with participants' desires to help the dogs, their expressions also conveyed a sense of wanting to provide help to dogs that are most in need of help. P3 described prioritizing rescue efforts for puppies and senior dogs due to them being more susceptible to diseases and welfare concerns.

*We [rescue] all dogs but when it comes to puppies or really old seniors we'll take those in a heartbeat because no senior should live a life at a shelter or go down because they're old, and puppies have their whole life ahead of them. [. . .] I think the more vulnerable are the ones you're trying to get out as soon as possible because they're at higher risk of getting sick or dying. (P3)*

Salient amongst the participants was also the need to help regions affected by ongoing global political issues. For example, P3 and P9 were coordinating rescue efforts in Afghanistan and in Ukraine, respectively. At the time of the interview, both of these regions were in the midst of a geopolitical crisis, with Afghanistan experiencing large-scale abandonment of companion animals due to the sudden end to the US-Afghanistan conflict [56], and Ukraine similarly suffering from the displacement of people and their companion animals due to the Russia-Ukraine conflict [57, 58]. These findings suggest that participants view their work as a moral good, and give support to the suggestions made by the Canadian National Canine Importation Working Group, where they stated that most dog importers are "trying to do the right thing" [10].

**Help (Canadian) adopter.**   Participants also identified the humanitarian benefits of providing a dog to an owner as a key motivator as to why they chose to be involved in international dog rescue. For example, some participants stated that a sizeable portion of adoption requests were from elderly adopters in their *"70, 80, even in their 90s looking for a [senior] dog." (P4)* Providing this specific type of adopters with dogs was viewed to be a win-win, resulting in a positive outcome for both the dog and the adopter. Participants described that *"it feels often*

*that you're actually not only helping the dog, but you're helping the person." (P1)* These added benefits strengthened participants' reasoning behind their involvement in international dog rescue efforts.

Some participants also shared their insight of rescue efforts from the supply and demand perspective. For instance, P9 argued that there is an undersupply of adoptable dogs in Ontario, but an abundance of adoptable dogs from other countries. In essence, the existence of international dog rescue operations aided both Canadian adopters and their international partners.

*Regionally funded [Ontario] humane societies are importing dogs [. . .] because there's not that many dogs here in Ontario. [. . .] I went to [. . .] the [local] Humane Society the other day because they donated some food for us. They have four dogs and they're all in fosters, right? So, the demand is greater than the offer here in Ontario. So, I think it's a combination of two. There's a lot of people that want to adopt dogs [. . .] And then there's so many dogs in need [. . .] Every day I get, I don't know 30, 40, 50? [requests] about dogs that are [abroad] that are going to get killed right now because they have no options. (P9)*

P5's motive for mitigating the undersupply of dogs was multifold. This participant echoed the additive benefits of saving a dog's life and the therapeutic benefits for elderly adopters but also believed that by facilitating the placement of international rescue dogs, they were preventing Canadians from acquiring dogs from less reputable sources (i.e., puppy mills). P5 also took issue with the unthoughtful breeding of dogs despite communities across the world euthanizing large quantities of dogs due to overpopulation.

*I think the fact that there's such a desire for animals here [in Canada. . .] and how few [dogs] there are here for adoption, right? So, people are desperate to get dogs and they're buying dogs from disreputable breeders and other places where animals are being mistreated. I mean if you can't meet the demand, people are going to go and buy from backyard breeders and that's really what I don't want. There's something morally reprehensible about euthanizing millions of animals and breeding the same amount at the same time. (P5)*

Previous studies suggest that low supply and high demand for dogs drives online puppy sales, which may also encourage fraudulent sales and scams involving imported dogs due to the inflation of dog market prices [27, 28]. Moreover, the existence of large numbers of international rescue organizations, itself, may suggest that there is high demand for adoptable dogs in Canada—as animal relocation programs address resource disparities by moving animals from areas of high rescue need to areas of high demand [59]. However, there is currently limited information on population-level supply and demand for dogs and further research is needed to corroborate the worldview of our participants regarding a potential shortage of dogs in Canada.

**Barrier.** Over half of the participants engaged in both local and international dog rescue. However, participants identified barriers that made the development of local rescue partnerships more challenging, but these barriers seemed absent when developing international partnerships. P1 highlighted how these barriers prevented their rescue organization from helping local communities. They expressed, in a closing statement, that they were willing to work with local partners but faced logistical challenges that prevented them from accessing remote Canadian communities.

*We would certainly be happy to rescue dogs from the [reserves] in Canada where there isn't so much spay and neuter, so they probably need homes. But we can't get up there and we can't get [the dogs] here. (P1)*

Other participants experienced barriers in the form of resistance from local rescue partners. Many participants similarly described facing pushback, or even criticisms, from other dog rescue organizations. P6, the founder of a BC-based rescue organization recalled facing resistance from local shelters during their early networking stages.

*So Canadian areas we have rescued from mainly are Manitoba because we have built connections there with some very dire needs as well. So, there is Canadian need. It's just not in our face and as apparent as some of these [international] areas, right? So, in some ways I would've loved to work with local Canadian shelters but, one, [they] didn't want to work with a small [rescue organization] they've never heard of. So, my drive to help dogs in need was there so I started networking. (P6)*

P2 described a mixture of societal and logistical challenges that factored into the difficulty of developing local rescue partnerships. They described that the abundance and the variety in international partners eased the identification of compatible rescue partners. In contrast, veterinary procedures were more costly, and local dogs were less accessible, which acted as barriers to developing local rescue partnerships.

*I would say we're predominantly international, and I think the reason behind that which is very interesting, it's because it's easy. It's so easy to. . . and I mean that is laughable because I would need to drink alcohol to say that, because it's not [actually easy, but only relatively easier]. What's easy is that, dogs are accessible and in need in other countries, whereas [for local partnerships, it's] like building relationships with local people and doing so in a way that's really sensitive and beautiful is a lot more difficult for some reason locally. Whereas internationally again, all these dogs are in abundance you know, there's so much variety whereas, local work is. . . It's hard. And also the costs are so different. You work with a dog in Dominican [Republic], like it costs me 30 bucks to neuter a dog, like it costs me $500 to neuter a dog here. The Americanization of dog welfare. . . Well actually I wouldn't even say dog welfare; the pet industry is whack. So, you're dealing with the consequences of that when you deal with local work. (P2)*

In contrast to local rescue partnerships, participants faced fewer obstacles when connecting with and expanding their international rescue network. In fact, participants expressed a sense of effortlessness for growing their international network. Participants described that "*[international] rescue connections kind of just evolve [. . . through] social media." (P2)* Participants recalled similar experiences, where they received a flood of rescue requests from international communities following successful rescue events.

*As I connected in with more groups and became part of [an international group] as well, that's when [the requests] really pours out. As soon as another international group posts [on social media] about how much you have helped them, it spreads. I mean you get messages from Korea, China, India, Pakistan, Mexico, Brazil, you know, they all just start coming out of the woodwork. (P3)*

The barriers for developing local rescue partnerships may be partially explained by the unique geographical and historical contexts of rural Canadian communities. Boey (2017), interviewed members from three First Nations communities in the Northern Interior of BC and their animal welfare partners to identify factors contributing to free-roaming dog population challenges faced by members of rural Canadian communities. They reported that the

more rural a community, the more common free-roaming dog related issues were, as the communities were socially and geographically isolated, which limited their access to information, technologies, and veterinary services. Similarly, Baker and colleagues (2020) reported that many Northern Canadian Indigenous communities lack access to veterinary services due to their remoteness, but access to these areas are also limited depending on the season due to its harsh climate [9].

The development of rescue partnerships with Canadian Indigenous communities may also be complicated due to recent and historical experiences of mistreatment. Boey (2017) reported that some Indigenous communities expressed mistrust of outsiders, particularly with animal welfare groups and cruelty investigators; some community members developed a negative impression of rescuers when their dogs were removed without their permission [8]. Taken together, local dog rescue efforts may be particularly challenging for individual dog rescue organizations due to logistical, financial, and societal barriers. As a result, potential rescue efforts may be being diverted to international communities. Local dog rescue needs may be better supported by assisting dog rescue organizations develop partnerships with local communities.

## Challenge

The theme *Challenge* captured aspects of international dog rescue work that participants found most strenuous. Four sub-themes were generated that best captured challenges commonly experienced by participants: Adopter expectation; Emotional stress; Rescue stigma; and Finances.

**Adopter expectation.** Participants identified interactions with adopters as one of the major challenges of international dog rescue work, particularly with the management of adopter's expectations in relation to the behaviour of the rescue dog and caretaking needs. P9 described this mismatch in expectations and in ownership responsibilities between them and their adopters as one of the biggest challenges they faced in their line of work.

> *I think the biggest challenge. . . Yeah, I'll say two and it comes down to people. [. . .] But it's responsible dog ownership. [. . .] We have these dogs fully vetted. I mean this example that I gave you about Sweetie, the dog, the Great Pyrenees [that was rescued from a road accident]. She had a family when she came here. And [after] she went to the family, [they] called us within a week and said "I don't want to be changing the bandages [. . .] so I want to take her to the vet to change the bandages. That's $100 every time I go to the vet. Can you pay for that?" And we're like, you adopted this dog. So, either return the dog to us and we will take care of her medical needs, or you need to be responsible for that. We cannot be paying for the dogs in perpetuity, like that's not how it works, right? (P9)*

Some work has reported that dog owners have higher expectations for their animal's behaviour than did cat owners [60]. Dogs expressing so-called "negative behaviours" has been cited as one of the most common reasons for why owners chose to relinquish their pet [60–62]. Furthermore, Powell and colleagues (2022) reported that adopters that chose to return their dog to shelters following adoption had higher expectations for the behaviour of their dog and the human-dog bond compared to adopters who decided to keep their dog [63]. Consequently, signs of high expectations by prospective adopters were generally not preferred by the participants when they discussed their respective adopter screening processes. In the voice of P1:

> *As part of our screening, we're looking for people who has that understanding that this is not going to be a North American dog who's lived in a house and is comfortable with all the same*

*things. You need to understand that it will take the dog a little bit longer than other dogs to adapt. [. . .] So that is something we look for, but it's always hard to be sure. Some people will tell you all the right things [. . .] and then when their dog is not reacting perfectly after the first week, they want to return it. (P1)*

Participants' great emphasis on adopter expectations may reflect their inherent empathy for the rescue dogs that flow through their organization. Many participants expressed having a strong emotional connection with the animals and, thus, were personally invested in adoption placement decisions. Their perceived role appeared to represent a guardianship role that required them to ensure the best outcomes for their rescue dogs. As P7 expressed, *"I get very attached to the dogs and I get really, really fussy about where they are going." (P7)* This protective role of the participants may have resulted in feelings of frustration when the adopters did not display similar levels of commitment towards dog ownership responsibilities.

Previous studies support the participants' concern for high adopter expectations, as mismatched owner expectations and actual caretaking efforts are potential risk factors for the relinquishment of dogs. Patronek and colleagues (1996) reported that roughly a third of owners that relinquished their dog expressed that the caretaking efforts were greater than they had expected [64]. Similarly, Diesel and colleagues (2008) reported that owners that underestimated the effort and work required in owning a dog were 9.9 times more likely to relinquish their dog compared to owners who had overestimated the caretaking efforts [61]. Taken together, the management of prospective adopter's knowledge and expectations for actual caretaking efforts associated with dog ownership seem to be an ongoing challenge among dog providers. Further research on ways to better equip prospective adopters with knowledge on dog caretaking efforts may be beneficial.

**Emotional stress.**   The emotional stress of dog rescue was also expressed by participants as a major challenge. This finding is not surprising given the ubiquity of poor mental health outcomes seen among individuals in caregiving professions, including in nurses [65], social workers [66], veterinarians [67], and animal shelter staff [68, 69]. Interestingly, there was disparity in how participants labelled the stress they experienced through dog rescue. P3 described their experience as post-traumatic stress disorder (PTSD):

*PTSD is a huge reality. A lot of people, SPCA, cruelty officers and others have to quit due to that, and I've experienced it myself many times and I think even the strongest of minds break. I [would be lying] if I [said] I haven't broken down a few times (P3)*

P6 described their personal struggle with combatting the emotional distress they developed through dog rescue, which they identified as compassion fatigue.

*Compassion fatigue, I didn't know what that was until I developed [it]. Rescue has made me develop anxiety. My Facebook feed, I've had to unfollow a lot of things [. . .] It's horrendous, what you see [online. . .] So, you know, it would anger me in my life because I would see my family continuing with Thanksgiving and I'm here on my phone seeing these horrific things. And then you kind of feel alone because they're like "oh you've got too many dogs in your house", and they're not seeing what you see, so you kind of alienate yourself [. . .] So that was a learning curve that I had to [. . .] learn to shut off. (P6)*

Variability in the definitions and conceptualizations of the various types of mental distress can also be found within the scientific literature [70]. There is particular variability surrounding the use of secondary traumatic stress and compassion fatigue, which are at times used

interchangeably [71], or as a sub-component of one another [70]. Nonetheless, regardless of whether it is compassion fatigue or secondary traumatic stress (or both), emotional distress appears prevalent among individuals in professions that provide care to patients, particularly in situations where there is continuous exposure to desperate situations that require care and empathy [68, 71, 72]. The findings of the current study suggest these challenges may also be prevalent among dog rescue members.

P6 described how a lack of understanding from close family members led to a feeling of alienation, which seemed to amplify their struggle with anxiety. In line with their experience, studies also suggest that lack of social support may increase worker susceptibility to stress [68]. Providing more social support is suggested to "buffer" the emotional burdens of work [68]. This was recognized by one of the participants (P8), who facilitated strong social support between staff members as well as limiting their dog rescue numbers as strategies to combat the emotional burdens of work.

*I try to get the [rescue] board [members] in a better headspace. We try to let things slide [. . .] So, it's more working on ourselves; to be able to cope with the stresses that come with [dog rescue. . .] So, we have meetings where we decompress [. . .] We'll go out to dinners or lunches together and we kind of have that bond as friends as well [. . .] To create a [rescue limit]. And that was healthy for us, because we were pulling everything and anything, and it was becoming very stressful. So now we have a number that we have to stick to, and that's kind of cooled everything down for us. (P8)*

Secondary traumatic stress and compassion fatigue is also often coincided with mental, emotional, physical, or physiological burnout [65, 68, 72]. Interestingly, some participants in the current study echoed the prevalence of burnout among dog rescue volunteers and colleagues. As a result of the high rate of burnout, participants recurrently expressed being understaffed and experiencing difficulties recruiting long-term members. *"[There are challenges finding] support, as far as volunteers or employees, especially ones that really stick around because [of the defeating nature of] this type of work." (P3)* While no previous research has examined occupational health conditions of dog rescuers, it is plausible that they face similar emotional burdens to those experienced by other caregiving professionals. The presented findings reinforce the importance of strong social support and workload reduction in caregiving professions to offset the inherent mental burdens of work.

**Rescue stigma.** Participants also described the stigma associated with international dog rescue as a major challenge. Participants recurrently described feeling misunderstood and experiencing premature judgement by other members of the community despite their dedicated dog rescue efforts. Participants described facing stigma and judgement from a wide demographic, including the general public, members of border services, veterinarians, and even other dog rescue organizations. P2 shared their perspective on the stigma surrounding international dog rescue.

*I think stigma is big [. . .] that's stigma [from] the community, what they think about rescue dogs [. . .] It's very rare that I meet someone at customs that I have a good rapport with. You know people in industry like customs and border security just don't have our backs most of the time [. . .] In just so many ways the stereotype or stigma of dogs from other countries bringing in diseases. [. . .] I feel like that's a really harmful way of understanding international rescue and demonizing an industry that is meant to support so many animals. (P2)*

Participants pointed to a widespread notion among members of the community that believe international dog rescue organizations to be a danger to public health as they engage in the

relocation of dogs potentially carrying diseases. Interestingly, participants identified other international dog rescue organizations as a catalyst for the rampant stigma surrounding international dog rescue, which participants referred to as *"well-meaning, not well-educated [rescues]." (P6)* There was a sense of uneasiness among participants for these organizations as they presented a reputational risk for responsible dog rescue organizations through engaging in careless importation practices.

> *Another challenge would be other rescues that ruin the name of rescues, right? Just like dog training, [dog] rescue is unregulated. So, you have a wide spectrum of how people are operating in rescue [. . .] We like to hold ourselves to the higher standards and do the best we can for the dog [. . .] we're very critical with how we work. [But it's] a challenge when a rescue [. . . receives] bad publicity [. . .] I think these people [are] well-meaning, but [. . . they] present challenges to us because we are constantly being compared [to them]. (P6)*

P5 suggested legislative changes to increase dog import standards and to lessen the stigma surrounding international dog rescue. It should be noted that the interviews undertaken for this study were conducted prior to the ban of commercial dog imports from non-rabies free countries introduced in Sep. 2022 [73], meaning that the only requirement for the importation of "commercial" dogs, the classification under which rescue dogs cross the border into Canada, over three months of age was a proof rabies vaccination or a veterinary certificate verifying that the dog originated from a rabies-free country [15]. P5 conveyed in their expression that these standards were too minimal.

> *I mean there are lots of rescues out there who are not great. They're well-meaning and lovely people that they are bringing animals but are dangerous and have diseases. And we need to stop that because [. . .] it's a reputational risk for all rescues even those who perhaps go above and beyond. So, for me, the biggest and most important thing, and of course as an importer everybody thinks you don't want regulation, I want more regulation. I really do. I want all dogs to be tested for parasites before they come to Canada. I want all dogs tested for heartworm. All dogs should be going through a temperament test process. I think it's easy for us to suggest the importation of dogs is inappropriate when people aren't doing their job properly. (P5)*

While poorly coordinated international relocation of dogs can present a major risk to public and animal health, risk of disease introduction can be effectively controlled with enforcement of medical requirements and compliance from the importers [74, 75]. Moreover, participants expressed a general lack of support for their efforts, frequently attributing this to the negative stigma associated with international dog rescue. Future research should consider conducting a focus group study with representatives of each stakeholder (i.e., veterinarians, general public, and international and local dog rescue organizations) to explore needs and expectations from each member. Further exploration of the tension between the stakeholders of dog importation may help to understand important considerations when developing future dog import policies that balances veterinary concerns while allowing dog rescue organizations to support international communities.

**Finances.**   Managing the financial costs of dog rescue was a common challenge among participants. P3 described finances as a key component of coordinating international dog rescue, as many of the necessary procedures were costly and international rescue operations were not possible without meeting these costs. P3 commented on the scale of the financial difficulties dog rescue organizations are facing, but implying that these difficulties are unavoidable as private rescue groups are attempting to tackle a government-scale issue.

*Finances. It's [a] huge [challenge]. It's probably the most [significant challenge] for any rescue. If we were all millionaires, the world would be a different place, right? If we had endless funding, my god. We're independent people basically taking care of a global government issue. Each countries' overpopulation is their issue, right? But they don't address it, so we do. But number one [challenge] for any rescue around the world is finances. Nothing happens without money [. . .] You can have all of the love and compassion in the world, which is great. I mean you need that first and foremost [. . .] But from [the] standpoint [of challenges], it's finances. I can't fly a dog, I can't bring it to the vet, I can't pay for its food [. . .] without money.* (P3)

The participants also openly revealed the financial models of their rescue organizations, which appeared to have many similarities, although expressions of the extent to which finances were a challenge differed between participants. For example, P7 expressed a lack of resources to support having a large foster-parent pool. Their descriptions of finances carried a sense of constraint for their rescue organization as they were unable to sustain having a large pool of foster parents, who are volunteers providing a temporary placement for the dog, due to limited funds for dog food and similar expenses.

*Like my husband says, "as long as [dog rescue] doesn't cost us money". And so, I can't, you know, look [to] having a lot of fosters and stuff like that where you have to pay for food and things like that, I can't afford to do that.* (P7)

In contrast, P8 was a veteran in the dog business having over two decades of experience, and unlike many of the less established participants, P8 did not cite finances as a major difficulty of their work. This participant seemed to rely on an extensive network of personal connections, including businesses outside of dog rescue, which facilitated their work, such as having a seemingly unlimited supply of dog food and other products.

*A little bit [of cost] on transport, mostly on vetting [. . .] As far of supplies, we don't, I mean I can get pounds and pounds of dog food for free and all that kind of jazz. I also run a board in kennel, and a daycare, and I [also] train [dogs professionally] so I have supplies coming out of my ears. I think vetting would be the most costly, and then, we obviously have an adoption fee that goes right back to the rescue [partner] as well.* (P8)

While there has been a growing concern among the public and veterinary community that unscrupulous importers may disguise themselves as reputable dog rescue organizations for monetary gain [10, 21], there were no mention of illegal dog importation activities during the interviews. Nonetheless, further investigation into the prevalence of illegal dog importation activities in North America is recommended.

Overall, participants expressed being financially limited in their rescue efforts. However, as revealed in the findings of the current study, some rescue organizations may be well-resourced in goods where other rescue organizations are lacking. This further suggests that collaboration between dog rescue organizations (as well as with other animal sheltering groups) and improved distribution of surplus resources may lead to even greater animal welfare outcomes.

### Duty

Lastly, the theme *Duty* captured participants' beliefs on responsible practices that should be followed by organizations that engage in international dog rescue efforts. Duty consisted of three sub-themes, which described the roles of international rescue organizations in facilitating

best outcomes for their relevant demographic sectors: Owner-dog relationship, Canadian society, and Rescue partners.

**Owner-dog relationship.** Throughout the interview sessions, participants expressed strong support for facilitating a 'good match' between the adopter and the rescue dog. Three concepts emerged as good practices of rescue organizations to promote compatibility between the adopter and their rescue dog: Dog assessment, Honesty, and Eternal safety net.

**Dog assessment.** Participants identified that the rescue organization's role in securing a good match began with the critical assessment of the dog's temperament as well as the determination of additional needs the dog may require to successfully adjust to their new, and often drastically different, lifestyle. International rescue organizations are unable to directly assess the behaviour of their own rescue dogs and must instead rely on their international partners to conduct these assessments, *"most of these groups, [whom, you] will never get the chance to meet." (P3)* To ensure that all dogs met their rescue standards, participants used video evaluation as part of their temperament assessment.

> *[Our rescue partners] have to show us videos of the dog interacting with people, with other dogs, if the dog is suitable with cats and children, then we can promote that dog as being able to live with them but only if we've seen a video showing they can interact with a cat or a child. (P1)*

Not surprisingly, participants expressed a degree of uncertainty when determining the dog's behaviours due to drastically different conditions of the dog's backgrounds to their new environments. This may explain the high rate of behavioural assistance adopters sought following the adoption of international rescue dogs [11]. P3 described that rescue dogs may appear to behave well but may show different behaviours after moving to a new environment.

> *Sometimes dogs can be in a shelter with a hundred dogs and they're acting perfectly fine, but then when you bring them here, they're super reactive. But that's also because [the dog is] forced to act proper when [they are] put into a [high intensity] shelter. [. . .] So, they tend to act right really quickly, because they're forced to. (P3)*

Within the temperament assessment, dog's sociability, particularly with other dogs, was recurrently emphasized as an important criterion. P9 expressed that their primary adopters are "families", in which the household environment requires dogs that are sociable with people and other dogs. This finding is in agreement with previous work reporting that the majority of adoptive households for international rescue dogs were families [11].

> *Our market is families. People want a dog that can go on walks [and] hiking, or be there with the family [. . .] Our criteria is pretty simple. It has to be a dog that is dog and people friendly. Right? Because people like to walk their dogs. People don't want the dog that is going to drag them down the street to go and attack [someone], so dog and people friendly. (P9)*

Participants also placed a strong emphasis for the evaluation of dog's level of aggression. However, there was a disparity among participants in their availability for rescuing dogs that display aggressive behaviours. Several participants immediately disqualified dogs with aggressive behaviours due to a lack of qualifications or resources *"we can't take [aggressive dogs] because we're not qualified to handle that." (P1)* In contrast, P3, and a few other participants, mentioned taking dogs regardless of the severity of their behavioural issues. In these latter cases the rescue organizations spoke about having animal behaviour professionals on-hand that allowed them to assist dogs requiring major rehabilitative work.

*We're pretty open here because we have a lot of experience with rehabilitating dogs with all sorts of behavioral issues. [. . .] A lot of rescues will have their specifics understandably, and that doesn't make us any better, but we try to stay pretty open. [. . .] but of course, if a dog shows up here and is literally trying to attack humans, that's a serious problem. And I don't just mean like, it's scared and I cornered it to pet it and it nipped me. I mean like it saw a human, and like went. [. . .] But most rescues to be honest, would never send a dog like that over. (P3)*

Studies have shown that behaviour in dogs plays a major role in the owner-dog relationship. It has been suggested that people regard sociable dogs as more adoptable; an attribute also associated with additional characteristics such as higher intelligence and lower aggression [76]. While participants described taking careful precautions in assessing the behaviour of their rescue dogs, *"some slip by [. . .] because [international shelters are] basically flooded." (P6)* Norman and colleagues [11] reported that fewer than one percent of adopters of international rescue dogs sought behavioural assistance from their rescue organization, and instead sought advice from the internet, friends, and veterinarians, among other animal professionals. Readily accessible online educational resources for adopters to overcome post-adoption behavioural challenges, as well as preparedness from local animal professionals to assist adopters with internationally-rescued dogs may be beneficial.

**Honesty.**   Participants emphasized the need for full disclosure of the dog's information to their adopters; honesty with the required caretaking efforts was valued as a critical component for facilitating a good match. Participants' descriptions also conveyed a sense of bluntness to ensure that adopters understood the efforts required to care for a rescue dog. P4 recalled directly asking owners for their expectations of rescue dog behaviour to ensure their expectations were representative of actual rescue dog behaviours.

*You want to make sure the person who's getting the dog is aware of the [needs] as much as possible about the dog. You want to know the person is up for it. Sometimes you have to be pretty blunt with people. You know, [I've asked] "are you looking for a dog that's house trained?" A couple times I've had [adopters reply] "yes we want a housetrained dog." [To that, I replied] "Sorry but, I can't say they are house trained." (P4)*

Honesty also appeared in the context of financial costs. For instance, P7 stressed the importance of providing younger adopters with realistic awareness of the financial costs of owning a dog. In support of this, results of a recent survey reported that a major area of conflict between veterinarians and dog owners was the high financial costs of veterinary visits [77]. Financial reasons were also among the primary reasons owners reported for relinquishing their dog [78]. P7 disclosed details of their personal veterinary expenses to their younger adopters, and strongly urged the acquisition of pet insurance to avoid potential financial hardships.

*If it's young [adopters], I always make sure that they understand that the cost of having a dog is horrendous and can put you under very quickly if you don't get pet insurance. So, if they're young [and] just starting out, they got their first apartment, they got a job and now they feel like they can afford a dog, I tell them how much I spend on vet bills in a month [. . .] I say, "you need to get pet insurance", you know. That's almost a requirement as far as I'm concerned. (P7)*

The findings of the current study further emphasize the importance of honest communication of the realistic amount of time, effort, and cost involved with caring for a dog to

prospective adopters to facilitate successful adoption placements. Shelter members are also encouraged to communicate financial and behavioural efforts involved with dog ownership to prospective adopters as a way to minimize mismatched expectations and to reduce likelihood of dogs being returned to shelters [79].

**Eternal safety net.**   Rescue organizations' duties to the welfare of the rescue dog continued even after the adoption was finalized. Participants emphasized the provision of ongoing support to their adopters as a good practice in mitigating adopter hardships, and subsequently reducing the likelihood of adopters from choosing to rehome their dog. For instance, P7 provided support to their adopters through social media platforms, encouraging all adopters to connect with all rescue volunteer members that were involved in the rescue process.

> *We stay in touch with our adopters. We have Facebook chat groups with all of our adopters. And [the adopters] can ask questions anytime. We never close the group [. . .] We have myself, the adopters, and the team in [South] Korea, who are people who work at [the shelter], [shelter] volunteers, the volunteer [dog] walkers who have lots of information about the dog. If something goes wrong that they can [provide help]. (P7)*

P3 echoed this sentiment, although they expressed greater concern for the dog's welfare. They believed it was the rescue organization's duty to ensure the dog finds an adoptive household instead of being transferred between sheltering facilities.

> *We're always here for them. So, whether they message us 6 days later, 6 weeks, 6 years were always here for that dog to support them for any questions for anything that comes up. For the life of a dog, it always has to come back to [us]. So, if you adopt a dog and 14 years later you have to give it up, it has to come back to us. We just don't want our dogs thrown into shelters or online, bounced around. So that's kind of our oath to the dogs as well as the people, that we'll always take them back. (P3)*

*Eternal safety net* may be a possible explanation for why Norman et al. [11] found that most adopters of international rescue dogs returned their dog back the original rescue organization when choosing to rehome, although further research is needed to support this hypothesis. Participants shared the belief that major changes to a dog's life, such as a return, is a stressful experience and should be avoided as much as possible. As P4 expressed:

> *You definitely don't want to put any dog in any home because what's going to happen is [the dogs are] going to come back to you. And that's just more time consuming and problematic for everybody. And definitely not good for the dog because it's going through yet another change. It's not very helpful in their transition to a new home and their behaviour if they have any. (P4)*

The findings of the current study suggest that using specialized knowledge of individuals who came into contact with a given dog may be beneficial for supporting adopters in overcoming animal-related difficulties experienced during animal ownership, and that international dog rescue organizations should work to avoid having their rescued dogs be surrendered to local sheltering facilities.

**Canadian society.**   Two concepts emerged as primary duties of international rescue organizations to minimize the harm to Canadian society through rescue dog import: Thorough medical screening, and Specific rescue dog criteria.

**Thorough medical screening.**   The health status of the prospective rescue dogs was repeatedly mentioned as one of the most important factors to consider when rescuing a dog.

In the voice of P3, who described that thorough medical screening was not just for public health safety but also for the dog's welfare.

*I think number one [priority] in importing is health. So, has this dog been tested for rabies [titer], or for a tick-borne disease, or for bacterial infections, or for something that you know, we do or do not have [in Canada]. But regardless we don't want a dog to be sick when flying [. . .] So, you want to do your due diligence in making sure that you ask these rescues "I want to see all these medical reports. I want to see them [. . .] up to date. And if they're not, or you can't afford it, we'll pay for it. Go get this bloodwork done and show me the results. Go get this SNAP test done and show me the results." [. . .] And really, it's ultimately to make sure the dog's healthy, right? [. . .] We don't want to bring in a sick dog especially on a plane. That's never good for anyone. (P3)*

The emphasis on the dog's health is unsurprising given that a key concern associated with international dog rescue is the potential risk of zoonotic disease spread [10, 13, 17]. Participants described conducting medical procedures beyond the standard legislative requirements which, at the time of the interview pertained only to rabies vaccinations [73]. As P5 expressed:

*We pay over the odds for our vetting because there's certain processes that I want. So, for example, every one of my dogs get a fecal screening, right? I say, "do a full parasite test". Even if it comes back negative, they still get full worming processes. They also do heartworm testing, right? We also, I will vaccinate them for Leptospirosis which isn't part of their normal plan. And sometimes there's other things I want done to the dog. So, for example I just rescued a dog this week, large, flat-coated retriever. He's got a big gouge on his leg, now they would just leave that, right? But no, I want the vet to go in, I want him to, you know, clean that up, close the wound properly so that the dog doesn't have any bald spots or scarring, or suffering right? [. . .] So, there's a few things that we pay for extra. I'll pay for dental costs, ear cleaning, grooming, all of those kinds of things if necessary. (P5)*

Perhaps as a result of the dedicated efforts in screening, participants expressed that prevalence of disease incidents from international dog rescue was low. P2 stated, *"[We've] rescued 2,400 dogs [. . .] but I cannot think of a time where a disease was passed on to an animal in Canada from one of our rescue dogs." (P2)* There is currently little information surrounding actual dog import numbers and the risks associated with the importation of rescue dogs to Canada. Kwan [74] reported that risk of rabies introduction from imported dogs and cats can be controlled through proper legislative policies. This author modeled that, with the current import policies in place in Japan, it would take 49,444 years before a single case of rabies would occur. Implementation of similar quantitative risk assessment with Canadian importation policies may aid in the development of legislative policies that seek to mitigate any risk associated with rabies arising from dog importation while allowing for rescue organizations to aid international communities.

**Specific dog criteria.** Some participants revealed having additional considerations when selecting rescue dogs. P2 prioritized rescue efforts towards local dogs, and international rescue dogs were carefully selected not to displace aid from a local dog.

*So, we help dogs locally as well, and most dogs locally in Toronto, Ontario have behavioral needs or medical needs. So, when we're able to take on dogs with medical and behavioural needs we typically try to support local cases. So, when we're looking for dogs internationally were hoping not to, I guess take over the place of a dog in need in Canada. (P2)*

In a similar vein, P5 avoided importing certain breeds as they were already commonly found in Canadian animal shelters.

> We were very clear that we would not rescue dogs that were over-represented in the B.C. shelter system. So that includes pit-bulls, rottweilers, other large breed, huskies, those kinds of things. So, we made an agreement that we would not be rescuing those dogs. (P5)

Our participants illustrate that as members of international rescue organizations, they are mindful of their actions and strive not to impede local dog rescue efforts. This approach, if held by all organizations involved with international dog rescue, may suggest a potential misconception within the overall dog rescue discourse where those opposed to international dog rescue may believe these organizations help animals elsewhere but not in their own country [21]. Participants' expressions instead suggest that they are maximizing benefits to dogs on a global scale with their available resources. Differences between international and local dog rescue organizations may simply be a matter of degree, dependent simply on the source location of rescue dogs. Moreover, it is important to note that there are comparable risks of zoonotic disease spread and animal welfare concerns associated with long-distance transport for local dog rescue efforts [10].

**Rescue partner.** The final sub-theme captured the duty of responsible engagement with international partners of rescue organizations. Participants were critical of their own practices and engaged in sensitivity throughout their interactions with rescue partners. Sensitivity was particularly noticeable in the statement from P2, where they made sure that their rescue partners had control over the rescue interactions. P2 acknowledged that their rescue partners may have different values and lifestyles, and as such were careful not to impose their own standards and practices onto their rescue partners.

> *I think what's important to us is [that] our rescue partners lead our interactions with them. You know we work on reserves up north and we're there for them. We're not trying to barge in with our expectations and things. I've been in many opportunities where I could buy dogs off [of] people because I think they could live better in Toronto but that's not what it's about. It's about supporting dogs where they currently exist. We do spay and neuter clinics in other countries and it's hard to see dogs come in that aren't living a standard of [life] that is similar to what I might see as beneficial, but I think it's really about those small conversations or opportunities to share spaces with people and learn what they're going through [. . .] So, we just have to be sensitive to that in order to really imprint the space in a positive light I guess. (P2)*

P6 displayed sensitivity in a slightly different form, taking extreme care not to take advantage of the potential financial hardships experienced by their international partners.

> *We're rescuing from sometimes desperate situations. And depending on the desperation that the people living on what I call "on zero", sometimes they will cover all the costs. Right? When that starts happening though, people here start taking advantage unknowingly. So sometimes the [source] rescue will pay for a lot of the way, whoever you're working with. [. . .] But that's not fair also. So, depending on who we work with, I know our Mexican rescue, they take in dogs, we kind of work with them to then see what is the right fit dogs for us, and they cover most of the way and then we pay them a quote on quote "pull fee" plus their flight [cost] from Mexico. So that's our arrangement we have with them. (P6)*

Practicing sensitivity and self-reflection in animal care settings may help to avoid engagement in dog rescue efforts that could contribute to further social inequities between vulnerable

and more resilient groups. This may be particularly valuable for the Canadian sheltering context, in which animal rescues in Canada have historically perpetuated harmful colonial practices that have created challenging relationships with First Nation communities [8]. However, considerations for these inequities are also needed in the context of international rescue.

## Future research and limitations

The current study explored the perspectives and experiences of members of Canadian-based international dog rescue organizations–stakeholders intimately involved in dog importation activities in Canada, but to the best of our knowledge, not yet explored in the scientific literature. Within the theme *Motive*, participants suggested a potential shortage of adoptable dogs in local BC and ON shelters. Discussions for a potential shortage of dogs in North America have been taking place among companion animal experts [80], and the findings of the current study suggest that some communities in North America may be experiencing a shortage of adoptable dogs. In support of this sentiment, previous studies have also reported that adopters of international rescue dogs in the United Kingdom and Canada expressed the inability to acquire an adoptable dog through local sources as one of the primary reasons for their adoption decisions [11, 15]. However, it is also important to note that the Coronavirus disease (COVID-19) pandemic in 2020 had significant effects on animal shelter operations, including adoption rates and the availability of adoptable animals in North America [4, 27]. The effects of COVID-19 on rescue operations were also expressed by several participants, thus we remind the readers to be mindful of the unique timeline in which the current study took place when interpreting the results. Nevertheless, we encourage further research on the supply and demand of adoptable dogs in North America as it may provide meaningful information on owner's thoughts behind dog acquisition methods, and help assess the risks associated with international movement of dogs.

Additionally, participants of the current study identified logistical and societal barriers that made development of local dog rescue partnerships more challenging. Further research on attitudes held by members of local dog shelters, veterinarians, and communities experiencing free-roaming dog related issues in Canada may be helpful in identifying important expectations and needs of each stakeholder, and how to best facilitate local dog rescue efforts. Future researchers may want to consider the use of a focus group method, as it is a research technique effective at identifying nuanced attitudes held by each participating group [34, 81]. Furthermore, while the current study explored the perspectives of Canadian-based international dog rescue organizations, previous work has suggested that there are also dog rescue organizations based abroad that export dogs to Canada [10]. Further research on this group may also provide better understanding of dog import activities in Canada.

While the current study explored the unique perspectives of Canadian-based international dog rescue organizations, the findings presented in this study are qualitative in nature, thus are context-dependent [34]. While these findings provide rich information within the intended naturalistic settings, readers should be wary when extending these findings to other settings where there may be differences in historical, cultural, or geographical contexts. Moreover, the current study interviewed only nine individuals, severely limiting the generalizability of the presented findings.

An additional limitation is the potential subjectivity involved in qualitative research. While the current study has taken steps to mitigate potential concerns that arise from the inherent subjectivity, it is impossible to fully eliminate the subjectivity that arise from analysis of qualitative data. Readers are encouraged to access the interview transcripts when aiming for thorough understanding of the findings presented in this paper.

Finally, the current study could be affected by selection bias during the sampling procedure [82]. Self-selection bias in sampling occurs when there are differences between the obtained sample (i.e., respondents), and the "targeted but not obtained sample" (i.e., non-respondents) [82]. However, it is impossible for the researcher to know the magnitude of differences between the respondents and non-respondents, or if there was a difference at all [82]. The current study recruited participants through email contact, and full disclosure was made regarding the research topic and intent prior to commencing the research procedure. Participants had full control over their involvement with the current research. As such, the sampling procedure may have favored the recruitment of rescue organizations that engaged in socially acceptable practices to take part in the research.

## Conclusion

The current study explored the motives, challenges, and unique life experiences of members of Canadian-based international dog rescue organizations. Specifically, the current study found that members of international dog rescue organizations were driven by strong internal desires to provide animal and humanitarian aid. However, local dog rescue efforts were constrained by logistical and societal barriers unique to the Canadian context. Additionally, the current study revealed both similarities and differences in occupational experiences between international dog rescue organizations and other animal care professions. Specifically, difficulties with the emotional burden associated with caregiving professions was also present within dog rescue work. However, international dog rescue members also experienced additional challenges due to the stigma surrounding international dog rescue operations. The current study also reported participants' beliefs on good practices of international dog rescue organizations for achieving best outcomes for the owner-dog relationship, Canadian society, and rescue partners.

## Supporting information

**S1 File. Letter of initial contact.**
(DOCX)

**S2 File. Interview guide.**
(DOCX)

**S3 File. Interview transcripts.**
(DOCX)

## Acknowledgments

This study was not possible without the voluntary contribution from members of Canadian-based international dog rescue organizations. We thank all participants for offering their insight on international dog rescue efforts. We extend our gratitude to all members of the Applied Animal Biology Program at the University of British Columbia for their feedback and ongoing support.

## Author Contributions

**Conceptualization:** Kai Alain von Rentzell, Alexandra Protopopova.

**Data curation:** Kai Alain von Rentzell.

**Formal analysis:** Kai Alain von Rentzell, Alexandra Protopopova.

**Funding acquisition:** Alexandra Protopopova.

**Investigation:** Kai Alain von Rentzell.

**Methodology:** Kai Alain von Rentzell, Christiana Bratiotis, Alexandra Protopopova.

**Supervision:** Christiana Bratiotis, Alexandra Protopopova.

**Validation:** Kai Alain von Rentzell, Alexandra Protopopova.

**Writing – original draft:** Kai Alain von Rentzell.

**Writing – review & editing:** Kai Alain von Rentzell, Christiana Bratiotis, Alexandra Protopopova.

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
