## [Decision Letter · Decision Letter 0]

24 Oct 2023

PONE-D-23-18268"It's my calling", Canadian rescuer's motives and experiences for engaging in international dog rescue effortsPLOS ONE

Dear Dr. von Rentzell,

Thank you for submitting your manuscript to PLOS ONE. After careful consideration, we feel that it has merit but does not fully meet PLOS ONE’s publication criteria as it currently stands. Therefore, we invite you to submit a revised version of the manuscript that addresses the points raised during the review process.

ACADEMIC EDITOR:

Qualitative research studies should be reported in accordance to the Consolidated criteria for reporting qualitative research (COREQ) checklist or Standards for reporting qualitative research (SRQR) checklist. Further reporting guidelines can be found in the Equator Network's Guidelines for reporting qualitative research. Indicate the author's bias as higlighted by one of the reviewers and explain how the authors manage this bias in doing the study. The authors state in the supplementary 1 that "the goal of this research is to support foreign dog adoption by gaining a better understanding of the rescue process directly from the people involved in foreign dog rescue."

We look forward to receiving your revised manuscript.

Kind regards,

Harvie P. Portugaliza, D.V.M., Ph.D.

Academic Editor

PLOS ONE

Journal Requirements:

Additional Editor Comments (if provided):

Qualitative research studies should be reported in accordance to the Consolidated criteria for reporting qualitative research (COREQ) checklist or Standards for reporting qualitative research (SRQR) checklist. Further reporting guidelines can be found in the Equator Network's Guidelines for reporting qualitative research.

Reviewers' comments:

Reviewer's Responses to Questions

**Comments to the Author**

1. Is the manuscript technically sound, and do the data support the conclusions?

Reviewer #1: Yes

Reviewer #2: Partly

2. Has the statistical analysis been performed appropriately and rigorously? 

Reviewer #1: N/A

Reviewer #2: N/A

3. Have the authors made all data underlying the findings in their manuscript fully available?

Reviewer #1: Yes

Reviewer #2: Yes

4. Is the manuscript presented in an intelligible fashion and written in standard English?

Reviewer #1: Yes

Reviewer #2: Yes

5. Review Comments to the Author

Reviewer #1: This is a small, subjective study, heavily based on opinion. It is a clear and well-written article. The information is presented in a predominantly balanced fashion, however, the limitations need to be expanded upon to adequately capture the authors' implicit bias.

Limitations noted: There is no mention of the impact of the COVID pandemic on the volume of dogs available for foster/adoption in local shelters and it has been documented the pandemic did impact dog availability during the time the study was conducted. This is a potential bias or limitation. It is possible the low local shelter volume noted by the interviewees was due to the COVID pandemic and not an accurate representation of long-term dog availability in Canadian shelters.

The authors' support of international dog rescue/adoption should be noted as a bias of this evaluation. The bias is stated in the S1 supplement in which the authors state they are in support of foreign adoption. It was also noted that no questions would be asked which could paint the rescues in an unfavorable light. Many rescues (including rescues the authors cited in this manuscript) violate federal dog importation rules in Canada and the U.S. to fast-track the purchase/adoption of dogs placing people and other pets at risk of exposure to potential diseases. The questions asked by the interviewer are not likely to capture the true practices some of these rescue groups engage in due to the authors self-declared support for these groups' activities.

Line 911-913: suggest specifying post-operative analgesia was provided by the rescue. The way it's written could be interpreted as no analgesia was provided during the surgery which is inhumane, unethical, and against CVMA guidelines.

Reviewer #2: The manuscript of von Rentzell and colleagues is a qualitative study on the perspectives and experiences of stakeholders involved in dog import activities in Canada.

Here are some comments:

Abstract

L27-29: Is there scientific evidence to support this generalization?

Introduction

L71: Properly write the scientific name.

Methods

Please add the COREQ (COnsolidated criteria for REporting Qualitative research) Checklist.

L117: Is the number of respondents adequate (saturation)?

Results and Discussion:

Are there mechanisms in this study for triangulation by involving other stakeholders, such as policymakers and community members?

6. PLOS authors have the option to publish the peer review history of their article (what does this mean?). If published, this will include your full peer review and any attached files.

Reviewer #1: No

Reviewer #2: No

---

## [Author Response · Author response to Decision Letter 0]

16 Jan 2024

Reviewer #1: 

This is a small, subjective study, heavily based on opinion. It is a clear and well-written article. The information is presented in a predominantly balanced fashion, however, the limitations need to be expanded upon to adequately capture the authors' implicit bias.

 Thank you for your comment. We agree that qualitative research samples a small number of participants and relies on the experimenter as the “instrument” for the analysis, necessarily adding subjectivity into the analysis. The sample sizes are determined not through a power analysis (as is in quantitative research that uses inferential statistics), but through a process of “saturation”, whereby adding additional members would not be deemed to add much additional insights for the particular research question. The “subjectivity” or the positions of the authors are stated in the manuscript under the “positionality statement” in the Methods section- as is typically done in qualitative research methodologies. Additionally, all data are available and the methods are described in detail allowing others to draw their own conclusions based on their own world-view. The purpose of our qualitative research is not to assess generality and not to provide data on the “average” dog importer. Instead, the goal is to provide insights into the lived experiences of stakeholders involved in dog import. We hope the reviewer sees the benefit of qualitative research in this topic area. 

Limitations noted: 

There is no mention of the impact of the COVID pandemic on the volume of dogs available for foster/adoption in local shelters and it has been documented the pandemic did impact dog availability during the time the study was conducted. This is a potential bias or limitation. It is possible the low local shelter volume noted by the interviewees was due to the COVID pandemic and not an accurate representation of long-term dog availability in Canadian shelters.

 Thank you for your comment. The potential effect of COVID-19 on the participants’ experiences has been added on lines 1070-1078. We hope that this newly added section helps to remind the readers of the unique timeline in which the study was conducted.

The authors' support of international dog rescue/adoption should be noted as a bias of this evaluation. The bias is stated in the S1 supplement in which the authors state they are in support of foreign adoption. It was also noted that no questions would be asked which could paint the rescues in an unfavorable light. Many rescues (including rescues the authors cited in this manuscript) violate federal dog importation rules in Canada and the U.S. to fast-track the purchase/adoption of dogs placing people and other pets at risk of exposure to potential diseases. The questions asked by the interviewer are not likely to capture the true practices some of these rescue groups engage in due to the authors self-declared support for these groups' activities.

 Thank you for your comment. Regarding your concern for our biased support for foreign dog adoption, I believe the statement you are referring to is found in S1 Supplementary Materials - Letter of Initial Contact, in which we stated, “The goal of this research is to support foreign dog adoption by gaining a better understanding of the rescue process directly from the people involved in foreign dog rescue”. We would like to clarify that what we meant with “support foreign dog adoption”, was simply to explore this topic through scientific inquiry. More specifically, through a rich qualitative analysis of the experiences, perspectives, and ‘subjective realities’ of the people involved in this type of work.

 This is not to be confused with a biased illustration supporting international dog rescue. To our knowledge, no previous scientific work has explored this topic. By simply studying the international dog rescuers’ perspectives, this would “support” international dog rescue by providing the scientific community a better understanding rescue processes, and by sharing international dog rescuers’ unique experiences, perspectives, and attitudes. 

 Regarding your concern for the interview questions being designed not to illustrate dog rescues in an unfavorable light, we believe you are referring to the statement also found in S1 Supplementary Materials – Letter of Initial Contact, in which we wrote, “We will not ask questions of a personal nature or any specific questions that may disrepute your organization”. This statement was added for two reasons: One, ethical reason, to ensure that participants were made clear of the types of questions they were going to be asked, and of the researcher’s intentions for this study; and two, to help with the recruitment process, as many dog rescue members expressed distrust of researchers due to previous bad encounters. This statement is not to be confused with a biased depiction of dog rescue in the current study. 

Line 911-913: suggest specifying post-operative analgesia was provided by the rescue. The way it's written could be interpreted as no analgesia was provided during the surgery which is inhumane, unethical, and against CVMA guidelines.

 Thank you for your suggestion. The sentence “We also pay for analgesics when our dogs get spayed” has been removed from the quoted passage. We believe that this resolves potential ambiguity in the medical procedure while maintaining the overall message of the manuscript. 

Reviewer #2: The manuscript of von Rentzell and colleagues is a qualitative study on the perspectives and experiences of stakeholders involved in dog import activities in Canada.

Here are some comments:

Abstract

L27-29: Is there scientific evidence to support this generalization?

 Dear reviewer, thank you for your comment. Yes. This statement was based on the findings and reports of other scientific (and organizational) publications on the importation of dogs. I have attached a few references that support this general statement – that international rescue dogs are frequently relocated from areas with an abundance of free-roaming dogs to industrialized countries where free-roaming dogs are absent. Overall, we feel that this statement has sufficient scientific evidence to support it, and therefore, no changes were made to Lines 27-29. Please let us know if there is ambiguity in the way it is currently phrased.

References:

Doyle E, Gupta M, Spindel M, Dolan ED, Slater MR, Janeczko S. Impact of the timing of spay-neuter related to transport on disease rates in relocated dogs. Animals. 2020;10(4):1–11.

Dogs Trust. Puppy Smuggling: Puppies still paying as Government delays [Internet]. 2020. Available from: https://www.dogstrust.org.uk/puppy-smuggling/041220_advert report_puppy smuggling a4_v15.pdf

Munkeboe N, Lohse‐lind A, Sandøe P, Forkman B, Nielsen SS. Comparing behavioural problems in imported street dogs and domestically reared danish dogs—the views of dog owners and veterinarians. Animals. 2021;11(5):1–15.

Pieracci EG, Williams CE, Wallace RM, Kalapura CR, Brown CM. U.S. dog importations during the COVID-19 pandemic: Do we have an erupting problem? PLoS ONE [Internet]. 2021;16(9 September):1–12. Available from: http://dx.doi.org/10.1371/journal.pone.0254287

Norman C, Stavisky J, Westgarth C. Importing rescue dogs into the UK: Reasons, methods and welfare considerations. Veterinary Record. 2020;186(8):248.

Introduction

L71: Properly write the scientific name.

 Thank you for your comment. This section has been revised to address the correct scientific name of the zoonotic disease in humans. These names are not to be confused with the scientific names of the bacteria/parasite causing the disease.

Methods

Please add the COREQ (Consolidated criteria for Reporting Qualitative research) Checklist.

 Thank you for your suggestion. We have carefully considered the addition of COREQ (Consolidated criteria for Reporting Qualitative Research) Checklist to the current manuscript. However, we have decided not to add COREQ checklist as we felt that the merit of adding this checklist was too minimal. We only identified three items from the checklist (item 12, item 18, and item 23) that could’ve been expanded on further in the current manuscript, and we feel that the methodological description, decision trail, and other practices we have engaged in to enhance the qualitative research rigor of the current study do not warrant the addition of the COREQ Checklist.

 Furthermore, we believe that expanding on the three items from COREQ checklist do not significantly benefit the current manuscript. For example, in COREQ Checklist item 12, we could’ve added why non-participants chose not to participate in the current study. Expanding on the possible reasons would be purely speculative and would not add significant value to the presented findings. Similarly, the reason for why the current study is missing item 18, Repeat interview (and whether they were conducted), and item 23, member checking is simply due to the limited timeline and resources for conducting the current study. Please let us know if you feel strongly that the COREQ Checklist is necessary for the current manuscript.

L117: Is the number of respondents adequate (saturation)?

Thank you for your comment. Lines 268-272 now outline how data saturation was assessed for the current study. To make sure our reviewers are aware of our reasoning behind the assessment of data saturation, I also provide additional description here:

 Data saturation was assessed based on the work by Saunders et al. (2018) and Braun and Clarke (2021). Given that the current study is guided by a constructionist epistemology, the assessment for data saturation do not follow the positivist (or realist ontological) notion of “reaching a fixed-point in which there is no new data to be sought/found”. Rather, the constructionist epistemology assumes that meaning is created in the process of the researcher’s deep immersion with the data (hence, “constructionist”). On this basis, new meanings are always theoretically possible, and the researcher makes the situated judgement on when to stop coding (often after reaching the point in which further data collection becomes counter-productive). 

 For the current study, while we have made great efforts in recruiting as many participants as possible, recruitment proved to be a rather challenging task than we had initially anticipated. The study’s timeline and the effort involved in the recruitment process were also major constraints. Data collection was completed at 9 participants, as we judged that further recruitment and data collection would be unproductive. Nevertheless, we believe that 9 participants were sufficient for offering rich insight into international dog rescuer’s experiences and perspectives.

References:

Saunders B, Sim J, Kingstone T, Baker S, Waterfield J, Bartlam B, et al. Saturation in qualitative research: exploring its conceptualization and operationalization. Quality and Quantity. 2018;52(4):1893–907. 

Braun V, Clarke V. To saturate or not to saturate? Questioning data saturation as a useful concept for thematic analysis and sample-size rationales. Qualitative Research in Sport, Exercise and Health [Internet]. 2021;13(2):201–16. Available from: https://doi.org/10.1080/2159676X.2019.1704846

Results and Discussion:

Are there mechanisms in this study for triangulation by involving other stakeholders, such as policymakers and community members?

Thank you very much for your suggestion. We concur that involving other stakeholders, policymakers, or professionals in this discussion (perhaps through a focus group setting) would certainly be a fascinating approach to achieve triangulation. However, this would require a considerable amount of preparation and work to accomplish, which is beyond the scope and capabilities of the current study. We hope that future studies can attempt to do this, as we have recommended in lines 708-714.

---

## [Decision Letter · Decision Letter 1]

23 Feb 2024

"It's my calling", Canadian dog rescuers' motives and experiences for engaging in international dog rescue efforts

PONE-D-23-18268R1

Dear Dr. von Rentzell,

We’re pleased to inform you that your manuscript has been judged scientifically suitable for publication and will be formally accepted for publication once it meets all outstanding technical requirements.

Kind regards,

Harvie P. Portugaliza, D.V.M., Ph.D.

Academic Editor

PLOS ONE

Additional Editor Comments (optional):

Reviewers' comments:

Reviewer's Responses to Questions

**Comments to the Author**

1. If the authors have adequately addressed your comments raised in a previous round of review and you feel that this manuscript is now acceptable for publication, you may indicate that here to bypass the “Comments to the Author” section, enter your conflict of interest statement in the “Confidential to Editor” section, and submit your "Accept" recommendation.

Reviewer #1: All comments have been addressed

2. Is the manuscript technically sound, and do the data support the conclusions?

Reviewer #1: Yes

3. Has the statistical analysis been performed appropriately and rigorously? 

Reviewer #1: Yes

4. Have the authors made all data underlying the findings in their manuscript fully available?

Reviewer #1: Yes

5. Is the manuscript presented in an intelligible fashion and written in standard English?

Reviewer #1: Yes

6. Review Comments to the Author

Reviewer #1: (No Response)

7. PLOS authors have the option to publish the peer review history of their article (what does this mean?). If published, this will include your full peer review and any attached files.

Reviewer #1: No

---

## [Editor Report · Acceptance letter]

21 May 2024

PONE-D-23-18268R1 

PLOS ONE

Dear Dr. von Rentzell, 

I'm pleased to inform you that your manuscript has been deemed suitable for publication in PLOS ONE. Congratulations! Your manuscript is now being handed over to our production team.

Kind regards, 

on behalf of

Dr. Harvie P. Portugaliza 

Academic Editor

PLOS ONE